# Improving Purpose in Life in School Settings

**DOI:** 10.3390/ijerph20186772

**Published:** 2023-09-17

**Authors:** Chiara Ruini, Elisa Albieri, Fedra Ottolini, Francesca Vescovelli

**Affiliations:** 1Department for Life Quality Studies, University of Bologna, 47921 Rimini, Italy; 2Department of Psychology, University of Bologna, 47521 Cesena, Italyfrancesca.vescovelli@unibo.it (F.V.)

**Keywords:** purpose in life, eudaimonic wellbeing, school interventions, children, depression

## Abstract

*Background and aim:* The dimension of purpose in life (PiL) is one of the core features of eudaimonia and plays a crucial role in developmental settings. However, few studies have examined purpose in life in younger generations and verified if it is amenable to improvements following a wellbeing-promoting intervention. The aim of the present investigation is to explore correlates and predictors of purpose in life in school children and to test if it can be ameliorated after school-based wellbeing interventions. *Methods:* A total of 614 students were recruited in various schools in Northern Italy. Of these, 456 belonged to junior high and high schools and were randomly assigned to receive a protocol of School Well-Being Therapy (WBT) or a psychoeducational intervention (controls). A total of 158 students were enrolled in elementary schools and received a positive narrative intervention based on fairytales or were randomly assigned to controlled conditions. All students were assessed pre- and post- intervention with Ryff scales of eudaimonic wellbeing (short version) and with other self-report measures of anxiety, depression and somatization. Additionally, the Strengths and Difficulties Questionnaire (SDQ) was administered to their schoolteachers as observed–rated evaluation. *Results:* In both elementary and high schools, purpose in life after the intervention was predicted by initial depressive symptoms and by group assignment (positive interventions vs. controls). In older students, PiL was predicted by female gender and anxiety levels, while no specific strengths identified by teachers were associated with PiL. *Conclusions:* PiL plays an important and strategic role in developmental settings, where students can develop skills and capacities to set meaningful goals in life. Depressive symptoms and anxiety can be obstacles to developing PiL in students, while positive school-based interventions can promote this core dimension of eudaimonia.

## 1. Introduction

Over the past two decades, the growth of positive psychology research has yielded a renewed interest in youths’ wellbeing. Various authors have emphasized the need to promote wellbeing in this population, particularly after the detrimental consequences of the COVID-19 pandemic [1,2,3,4,5]. At the international level, psychologists, educators and policy makers have suggested introducing educational programs for promoting wellbeing as an integral part of the school curriculum [6,7,8,9,10].

However, previous research focused on happiness and hedonic wellbeing in children and adolescents, which were considered as key ingredients to optimal development [5,11]. Conversely, eudaimonic or existential wellbeing in the early stage of development has been neglected by existing investigations [7,12] because it was considered not easily understandable by younger children, in light of its abstractness and multidimensional nature [13,14]. Recent contributions, however, have modified these opinions [11,15,16]. For instance, Gillett-Swan [16] described how children aged 8–12 years were able to discuss wellbeing-related issues with deep, coherent and complex reasoning. Similarly, in a qualitative investigation, Ruini et al. [17] documented that elementary schoolchildren experienced moments of eudaimonic wellbeing in their daily lives. In fact, they reported activities with peers and with family members as common triggers for positive emotions. Indeed, this could be considered an indicator of positive interpersonal relationships, which is a core dimension of eudaimonic wellbeing. Additionally, another 20% of children in this qualitative study reported episodes of peak performance in sport and school activities, or the acquisition of new skills (i.e., learning to swim, ski, handcraft activities, etc.) as correlated with a sense of happiness and self-esteem. In this case, these statements can be understood as indicators of pursuing goals, another core dimension of eudaimonic wellbeing. Similarly, Tavernier and Willoughby [18] found that teenagers could ponder meaningful life experiences (i.e., turning points) and recognize them as crucial for their identity development. Adolescents’ wellbeing was correlated to this specific process of meaning making. Taken together, these investigations documented that children and adolescents are able to understand a sense of personal growth, self-esteem and goal achievement as being important contributors to their happiness. These findings are in line with previous literature that assigned to life purpose a significant role for facilitating the construction of a stable sense of identity in adolescence [19].

In fact, among all the different intercorrelated components of eudaimonia [12], purpose in life has been the most widely investigated dimension in youth and adolescents [16,19,20]. Early pioneering contributions in this field derived from traditional theories on cognitive, moral and identity development. For instance, in his cognitive development model, Piaget suggested that teenagers acquire the ability to reflect on themselves and beyond themselves, resulting in an increased interest in interpersonal relationships and in the surrounding world [21]. In a parallel process, cognitive development is associated with self-reflection and improved skills in moral reasoning during adolescence [22]. Similarly, Erikson [23] postulated that under optimal conditions, both purpose in life and a sense of one’s identity develop during adolescence. In fact, during adolescence, the resolution of identity crises is facilitated by the acquisition of a deeper sense of purpose in life [23]. In summary, classical psychological theories described young age as an ideal phase for nurturing purpose in life as a key ingredient for a positive transition to adulthood [19,20]. For instance, some pivotal studies provided evidence that acquiring purpose in life follows a developmental pattern. In early adolescence, the majority of students can describe their understanding of the concept of purpose in life when interviewed [16,24,25]. Similarly, in a qualitative investigation [26], the authors found that teenagers reported mature and complex conceptualizations of purpose in life: almost all participants could describe their ideas of ‘having a purpose in life’, and 68% indicated several related themes. Prosocial issues were reported by 26% of teenagers; after that, the role of religion (18%) and occupational and financial themes (17%) were also mentioned. Moreover, the majority of students considered purpose in life as an important contributor to their mental and emotional wellbeing.

A recent review on the role of purpose in life in school psychology and counselling [27] summarized the main findings of research from the past 20 years: teenagers with a greater sense of purpose also manifested increased life satisfaction and emotional wellbeing, a goal-directed cognitive style, a lower tendency towards poor psychosocial adjustment, and less risk-taking behaviors when compared to students with a lower sense of purpose [25,26,28]. Moreover, during adolescence, purpose in life was positively related to hope [29,30], since it promotes a more flexible sense of personal planning and agency. Finally, purpose in life in adolescence was found to be correlated to another core eudaimonic dimension: autonomy and identity exploration [19,23]. Conversely, the lack of purpose in life emerged as an indicator of psychopathology. Individuals reporting lower levels of purpose in life also lamented high levels of depressive symptoms [24,28,31].

Thus, purpose in life in youth can develop naturally as a consequence of identity maturation, and it could be considered an important indicator of eudaimonia, which is strongly associated with aspects of emotional and social wellbeing, and with a decreased risk of psychological distress [3,27].

However, in view of its important role in youth, purpose in life was considered an important target for teachers, coaches and educators, and specific psychoeducational interventions were created to promote/foster this characteristic in young generations [27,30]. One of the most known is the Positive Adolescent Training through Holistic Social Programs (PATHS), used in secondary schools, which is specifically tailored to promote students’ sense of purpose [32]. Other curricula and school programs are available, some of them also reflecting cultural differences. In Western educational settings (USA, Australia, UK), many school interventions teach students how to develop and pursue self-oriented life goals in order to achieve a greater sense of personal happiness and life satisfaction [5,7,10,30]. Other interventions follow a humanistic/existential perspective and are used more to promote spirituality and meaning in life, rather than goals and purpose [18,27,33].

Another possible path to promote purpose in life is by targeting eudaimonic wellbeing as whole, since it is composed of interrelated dimensions [12,34,35]. Pivotal work was conducted almost two decades ago by the School Well-Being Therapy program (School WBT) [36,37]. It relied on Ryff’s model of eudaimonic wellbeing [35], which includes six areas of positive functioning: autonomy, environmental mastery, purpose in life, personal growth, positive relations and self-acceptance. School WBT was tested in Northern Italy in various classes within middle and high schools [36,37]. This school intervention entailed four sessions delivered in the class during curricular teaching. It included an educational component on Ryff’s model of eudaimonic wellbeing followed by cognitive–behavioral techniques [36,37]. This school program was found to ameliorate anxiety and somatic symptoms and to enhance eudaimonic wellbeing in children. This school protocol was also applied in clinical settings, with the aim of treating children with emotional and behavioral disorders [38]. It resulted in improvements in children’s eudaimonic wellbeing and in a reduction in somatization [38].

More recently, the same group of investigators developed another school program to be implemented in elementary schoolchildren for promoting eudaimonic wellbeing in the early stages in life [39]. In this case, it entailed four sessions performed in the class, but it used narrative techniques and fairytale readings in order to facilitate children’s understanding of concepts such as autonomy, environmental mastery, positive relations and purpose in life. After this school intervention, children assigned to the narrative protocol reported improvements in their wellbeing and a reduction in depression, anxiety and somatization, when compared to the control condition. The intervention also fostered children’s creativity.

Taken together, these school programs have documented that the promotion of eudaimonic or existential wellbeing is feasible in youth, with various approaches and techniques. In these randomized controlled trials performed in schools, the wellbeing interventions (School WBT and Positive Narrative Intervention) were compared to controlled conditions and were found to be effective in improving most of the dimensions of eudaimonia, including purpose in life. Students assigned to the positive interventions reported significant pre-post improvements in their levels of eudaimonic wellbeing, together with decreased levels of anxiety, depression and somatization. Conversely, students assigned to the controlled conditions displayed a different pattern of change in their psychological dimensions: wellbeing dimensions did not improve at post-intervention assessment, and in some cases they also showed significant declines, particularly the subscales of purpose in life and personal growth (for further details, see [36,37,39]). Considering that few studies have examined the correlates of purpose in life in younger generations and their trajectories following a specific wellbeing-promoting intervention, with the present investigation we aim to analyze the effect of positive interventions on the specific dimension of purpose in life (post-intervention scores) in the overall sample of school students. Additionally, we aim to explore correlates and predictors of purpose in life in the total sample of school children, considering their baseline characteristics in terms of psychological distress (anxiety, depression, somatization) and of the evaluation (including pro-social behaviors) by their schoolteachers.

## 2. Materials and Methods

### 2.1. Sample

A total of 614 students (females = 358; 58%; age range = 8–17 years; mean age = 11.87) were recruited in various schools in Northern Italy. Of these, 456 belonged to middle (6–7th grade) and high schools (8–9th grade) and 158 students were enrolled in elementary schools (4–5th grade).

Recruitment procedures, study design and intervention protocols are described in details in previous publications [36,37,39]. The research protocols were approved by the academic IRB board. Each school enrolled in the study voluntarily participated and provided formal approval. A total of 3 elementary schools, 4 middle schools and 2 high schools were enrolled, with a total of 26 school classes. The schools recruited for the research project presented with similar sociodemographic characteristics: they were public schools settled in provincial towns in Northern Italy with low rates of ethnic minority groups and with a similar socio-economic status. All students involved in the class programs, their teachers and their parents provided written informed consent to participate in the study (primary inclusion criterion). Severe disability (such as neurodevelopmental disorder, autism spectrum disorder, visual impairments) or inability to speak and understand the Italian language were considered exclusion criteria, since these children could not be properly engaged in the activities or provide their individual contribution.

Study design: randomized controlled study. Randomization was performed at class level (not on single students). This means that in each school some classes received the eudaimonic wellbeing school programs, while other classes (same grade) acted as the control condition. In middle and high schools, students were randomized to receive a protocol of School WBT or a psychoeducative intervention (controls). In elementary schools, students were randomized to receive a positive narrative intervention based on fairytales, or they were assigned to a controlled condition, where fairytales were read and discussed with teachers as part of the traditional school curriculum (See Figure 1, Consort Diagram). In the present investigation, we analyze the effect of the intervention assignment (experimental vs. controlled condition) together with sociodemographic variables and indicators of psychological distress in predicting purpose in life in the sample of students.

### 2.2. Assessment

Students were assessed pre- and post- intervention with various self-reports for evaluating purpose in life and other indicators of psychological distress: anxiety, depression and somatization. Additionally, the Strengths and Difficulties Questionnaire (SDQ) was administered to their schoolteachers as an observed–rated evaluation. Assessment was performed by two psychologists not involved in the school interventions.

*Purpose in life* was assessed with the subscale of Ryff’s Psychological Well-Being Scales (PWB) (short version) [40,41,42]. It consists of 3 items (“I enjoy making plans for the future and working to make them a reality”, “My daily activities often seem trivial and unimportant to me”, “I have a sense of direction and purpose in life.”). The children answered on a 6-point Likert scale (1 = this is not my case; 6 = I totally agree). Negatively phrased items are reversed in the scoring procedure, so that higher scores represent higher levels of purpose in life. The scale score may range from 0 to 18. The total PWB scales were previously validated in an Italian population [40]. The psychometric properties for the full scale are good, with high interitem correlations and a good test–retest reliability, even though the short version of the PWB raised some criticisms for its weaker internal consistency and poor definitions of eudaimonic wellbeing dimensions [12,34]. However, this short version (18 items) was found to be the most suitable for younger populations [43]. PWB was used in several investigations with children and teenagers, both in clinical and school settings [37,41,44]. In the present study, Cronbach’s alpha for the scale was 0.14 for the total sample of students.

*Anxiety* was assessed with the Revised Children’s Manifest Anxiety Scale (RCMAS) [45]. It is a self-rating, 37-item questionnaire with yes/no questions for evaluating anxiety in young populations (age range = 8–19 years). The 37 items pertain to four scales: Physiological Anxiety (10 items), Worry/Oversensitivity (11 items), Social Concerns/Concentration (7 items) and the Lie Scale (9 items). A Total Anxiety score can be computed by adding the scores to the 28 anxiety items. Higher scores indicate greater levels of anxiety. RCMAS possesses good psychometric properties: high internal consistency, good test–retest reliability (α = 0.87) and predictive validity. In the sample of elementary schoolchildren, Cronbach’s alpha for the total scale was 0.75; in the sample of middle and high school students it was 0.69.

*Depression* was assessed with the Cognitive Triad Inventory for Children (CTI-C) [46] in elementary schoolchildren and with the Depression subscale of the Symptom Questionnaire [47] in middle and high school children. CTI-C is a 36-item, self-report questionnaire for the assessment of children and adolescents’ depression, following Beck’s cognitive triad model, since items are organized into three subscales (negative view of the Self, World and Future). Each one consists of 12 items and a total scale can be calculated by adding up the three subscales. Children answer using a 3-point format (yes/maybe/no). The questionnaire has a robust concurrent and internal validity (Cronbach’s alpha = 0.92). In the present study, Cronbach’s alpha for the total scale was 0.50 for elementary schoolchildren. The SQ-Depression (SQ-D) subscale is a 23-item self-rating scale with yes/no answers that contains items for assessing symptoms of depression and items for assessing wellbeing (i.e., happiness). Higher scores indicate greater depression. SQ has been extensively validated [47]. In the present study, Cronbach’s alpha for the scale was 0.60 for middle and high school students.

*Somatization* was assessed using the Children’s Somatization Inventory-Child Report Form (CSI) [48] in elementary schoolchildren and with the Somatization subscale of the Symptom Questionnaire [47] in middle and high school children. CSI is a questionnaire for evaluating the presence of somatic symptoms in children. CSI measures the severity of 35 somatic symptoms, which are computed through a 5-point scale (0 = never to 4 = always), referring to the last 2 weeks. The total score is calculated by adding the scores of each item/symptom and ranges from 0 to 140, with higher scores indicating higher levels of somatization. The tool has good psychometric properties, and it is positively correlated with other measures of anxiety and depression. In the present study, Cronbach’s alpha was 0.89 for elementary schoolchildren. The SQ-Somatization (SQ-S) subscale is a 23-item self-rating scale with yes/no answers that contains items for measuring symptoms of somatization and items for measuring physical wellbeing (i.e., no symptoms in any part of the body). Higher scores indicate greater somatization. SQ scales have been extensively validated [47]. In the present study, Cronbach’s alpha for the scale was 0.62 for middle/high school children.

All students were also evaluated before and after the intervention by their teachers (one teacher for each class involved in the study) using the *Strengths and Difficulties Questionnaire* (SDQ) [49]. The SDQ is a brief behavioral screening questionnaire that collects information from parents or teachers about their children’s strengths, as well as their difficulties. The SDQ is composed of twenty-four items divided into five subscales: Emotional symptoms, Conduct problems, Hyperactivity—inattention, Peer relationship problems and Prosocial behavior. It gives a score for each subscale or a total score, computed by summing the scores from all of the subscales [49]. The Cronbach alpha for the SDQ total scale was 0.55 in the elementary schoolchildren sample and 0.63 in the sample of middle/high school students.

### 2.3. Data Analysis

The sample characteristics were analyzed with descriptive statistics (frequencies, mean values, SD). Baseline differences among the classes were calculated using MANOVA, with class as a fixed factor, and purpose in life, CTI, SQ Depression and SQ Somatization, and CSI and R-CMAS Total Anxiety scale scores as dependent variables. Bonferroni post hoc tests were then applied to verify specific differences among the various classes.

In order to evaluate the predictors of purpose in life (post-intervention scores) in elementary schoolchildren and in middle and high school students, two multivariate linear regression analyses (method enter) were performed: one for the subsample of elementary schoolchildren and one for the subsample of middle and high school students. In both cases a four-step model was used, where socio demographic factors (age, gender) and intervention condition (wellbeing intervention vs. controls) (step 1); baseline teachers’ evaluations (SDQ scores) (step 2); baseline anxiety levels (RCMAS scores) (step 3); and baseline depressive and somatization symptoms (CTI and SQ-D scores; CSI and SQ-S scores) (step 4) were entered to test if they significantly predicted purpose in life. Two separate regressive models were used for different reasons: age trajectories are particularly relevant in developmental settings; and elementary schoolchildren may present relevant differences in their cognitive skills and in the manifestation of psychological distress (anxiety, depression and somatization) compared to older students [7,20,24,25] who were all teenagers. Moreover, we used different questionnaires to measure depression and somatization in the two subsamples. Baseline scores in purpose in life were not entered as predictors because previous investigations with a repeated measure design provided the pre-post analyses in wellbeing dimensions, including purpose in life [36,37,39]. For the present investigation, we focused on other possible predictors of purpose in life in the two cohorts of students, considering the baseline characteristics of the sample (levels of anxiety, depression, somatization and teachers ‘evaluations). Missing data were handled with an intent-to treat analyses.

All analyses were performed using Statistical Package for the Social Sciences (IBM-SPSS) version 28.

## 3. Results

Sample descriptive statistics are reported in Table 1. The baseline mean scores for purpose in life were 15.21 (SD = 2.97; score range 6–18) in elementary schoolchildren and 14.58 (SD = 3.12; score range 3–18) in middle and high school students. Purpose in life at post-intervention had a mean score of 15.17 (SD = 2.88; score range 4–18) in elementary schoolchildren and a mean score of 14.45 (SD = 3.12; score range 4–18) in middle and high school students. (See Table 1). A previous investigation [39] already documented that the nine classes within the elementary schools were homogeneous and did not present statistical differences at baseline assessment (for further details, see [39], Results section). The 17 classes in middle/high schools were compared using MANOVA, with class as a fixed factor and baseline score at purpose in life, anxiety, depression and somatization as dependent variables. A Bonferroni post hoc test was then applied. The only significant difference emerged on the anxiety scale (RCMS Total), where one middle school class presented a significant lower score (standard error = 1.45; *p =* 0.018), compared to another high school class. All other variables showed no significant differences due to class.

Two four-step regression models were calculated to predict purpose in life at post-intervention (dependent variable) in elementary schoolchildren and in middle and high school students, respectively. The regression model performed in elementary schoolchildren revealed that variables included in the fourth model explained 55% of the variance (F 12,145 = 5.248, *p* < 0.001). Particularly, lower depressive symptoms and being assigned to the wellbeing intervention predicted higher purpose in life scores (β = −0.399, *p* < 0.001, β = −0.174, *p* < 0.05, respectively) (Table 2).

The regression model performed in middle and high school students revealed that variables included in the fourth model explained 53.1% of the variance (F 12,443 = 14.469, *p* < 0.001). Particularly, female gender (β = −0.140, *p* < 0.05), lower scores in depression (β = −0.317, *p* < 0.001) and anxiety (β = −0.245, *p* < 0.001), and being assigned to the wellbeing intervention (β = −0.91, *p* < 0.05) predicted higher purpose in life scores. (Table 3).

## 4. Discussion

The purpose of this investigation was to explore the correlates of purpose in life in youth and to verify if it was amenable to change following a eudaimonic wellbeing intervention performed in schools. In previous randomized controlled investigations, it was documented that initial levels of eudaimonic wellbeing (in all its six dimensions) were significantly improved after a short school program, when compared to controlled conditions [36,37,39]. In Table 1, we report baseline and post-intervention levels of purpose in life for the overall sample of students, without differentiating between those assigned to the experimental conditions and those assigned to controls in elementary and middle/high schools. In the present investigation, we specifically aimed to further explore the trajectory of purpose in life, also considering other possible predictors, such as psychological distress and prosocial behaviors as evaluated by schoolteachers. The findings revealed that in both elementary, middle and high school students, purpose in life after the intervention was predicted by initial depressive symptoms and group assignment (wellbeing interventions vs. controlled conditions). In older students, purpose in life was also predicted by female gender and anxiety, while no specific strengths identified by teachers were associated with purpose in life. These results are in line with previous investigations that documented the inverse relationship between purpose in life and psychological distress, particularly depression [19,27,46,50]. Depression, in its core characteristics, entails a pessimistic view of the self, the world and the future [51]. On the other hand, purpose in life has been conceptualized as strongly associated with identity development [23,24] and with a sense of direction in life and goals to pursue [3,19]. Unsurprisingly, in many studies with adult and aging populations, purpose in life and depression were inversely correlated [52,53,54,55]. The findings of the present investigation also extend these relationships to young populations. In fact, the regression models performed in both elementary schoolchildren and middle/high school students revealed depressive symptoms as negative predictors of purpose in life; in older students, the other predictors were female gender and anxiety. Another recent meta-analysis documented the same predictors for hope (another positive domain similar to purpose in life for its future-oriented nature) in adolescence [56]. However, this meta-analysis found that hope was predicted more strongly by other positive characteristics, such as life satisfaction, positive affect and optimism. Unfortunately, we have not evaluated these aspects in our sample, and we agree with the authors of this meta-analysis that more research is needed on the correlates and predictors of future-oriented attitude in younger generations.

Our research, however, does not confirm the pro-social dimension of purpose in life [20,24,27]. Various authors have documented that purpose can motivate individuals to achieve goals for improving one’s life, but also (and more importantly) for making a difference to the lives of others in the same community [27]. We evaluated altruistic attitude in students with the SDQ subscale of “kind and helpful behaviors”, but the correlations were not significant in our regression models, neither for elementary schoolchildren, nor for older students. The SDQ scale was completed by students’ teachers; hence, it could be possible that their ratings did not capture the full spectrum of pro-social attitudes of our sample, or that observed evaluations in young populations do not always parallel self-evaluation [41]. Additionally, other culturally sensitive issues may have determined this lack of correlation. Our sample was mainly composed of Italian children/teenagers of similar socio-economic status (see details in previously published articles, [37,39,44]). With a more heterogeneous sample, it is possible that the results would have documented more pro-social behaviors in students with higher purpose in life, or belonging to more collectivistic cultures [27].

Finally, the present investigation documented that short school programs (four sessions) aimed at promoting eudaimonic wellbeing in students were able to significantly modify purpose in life when compared to control conditions. The contents of the programs and all the activities performed with students have been described in detail in previous investigations, where their efficacy was tested with controlled designs [36,37,39]. They referred to Ryff’s model of eudaimonic wellbeing [35], and the activities performed with students aimed to help them understand the concepts of personal growth, environmental mastery, self-acceptance, autonomy, positive relations and purpose in life, which can be difficult to assimilate for their abstractness [13,17]. In younger children, we used narrative strategies based on the reading, discussion and writing of fairytales with an emotional content. For their narrative plots, fairytales are particularly suitable to convey a sense of purpose and direction [57,58], and children were asked to write a new fairytale with a happy ending that consisted in solving one of their problems and/or achieving one of their goals. With middle and high school students, the activities included role playing and group discussion. For the session dedicated to purpose in life, we asked them to write their personal horoscope for the following year, to design and imagine their life in a ten-year period and to present themselves as adults to their classmate [36,37]. In both types of intervention, the focus was on future planning and goal achievements, but also on teaching their values for identity maturation. In the controlled conditions, these activities were not performed and, unsurprisingly, group assignment resulted as a significant predictor of purpose in life in our regression models for both elementary schoolchildren and older students.

The present research is limited for the characteristics of the sample: students were homogeneous, they had no particular physical problems or mental health issues (students with learning disabilities were excluded) and no ethnic minorities were involved. Another limitation concerns the fact that assessment was largely performed with self-report questionnaires, and only one teacher per class provided observed ratings of children. Thus, also obtaining data from students’ parents or other significant adults in their life would have provided a more reliable picture of their psychological characteristics [41]. Moreover, the main dependent variable (purpose in life) was only measured with three items from the short version of the Psychological Well-Being scale [42]. Some authors have criticized the psychometric properties of this scale, particularly in its short version [34,59,60]. From the original 20 items per scale version, three items were selected, with the aim of capturing the conceptual scope of the definitions. Even if authors documented that the shortened PWB version still correlated between 0.70 and 0.89 with the longer version, the internal consistency for each of the six subscales was low, ranging from 0.33 to 0.56 [47]. Also in the present research, Cronbach’s alpha was quite low (0.14), and this fact can be considered a major source of limitation. However, this is a necessary choice to make in order to maintain the multidimensionality of the construct of purpose in life [34]. As mentioned before, data on other positive dimensions of functioning (i.e., life satisfaction, positive affect, optimism) are lacking in the present investigations and might have resulted as significant predictors of purpose in life, as documented by another meta-analysis [56].

However, the findings of this study extend previous research [52,53,54,55] by confirming that depressive symptoms and anxiety can be obstacles to developing purpose in life in children and adolescents. More importantly, our findings suggest that short positive school-based interventions can promote this core dimension of eudaimonia in students. Even though the main focus of the present research is not the wellbeing interventions per se (already described in detail elsewhere), the present research suggests that purpose in life in children/adolescents could be easily promoted in school settings, with a particular beneficial effect in those children with pre-existing depressive or anxious symptoms.

## 5. Conclusions

Considering the important and strategic role of purpose in life in developmental settings [19], providing students the opportunity to acquire and develop skills to set themselves meaningful goals in life appears to be crucial for the future of younger generations. Positive psychology and positive education could provide effective tools and methodologies to pursue this important aim, while policy makers and educational systems should take the responsibility of making it possible at every school level.

## Figures and Tables

**Figure 1 ijerph-20-06772-f001:**
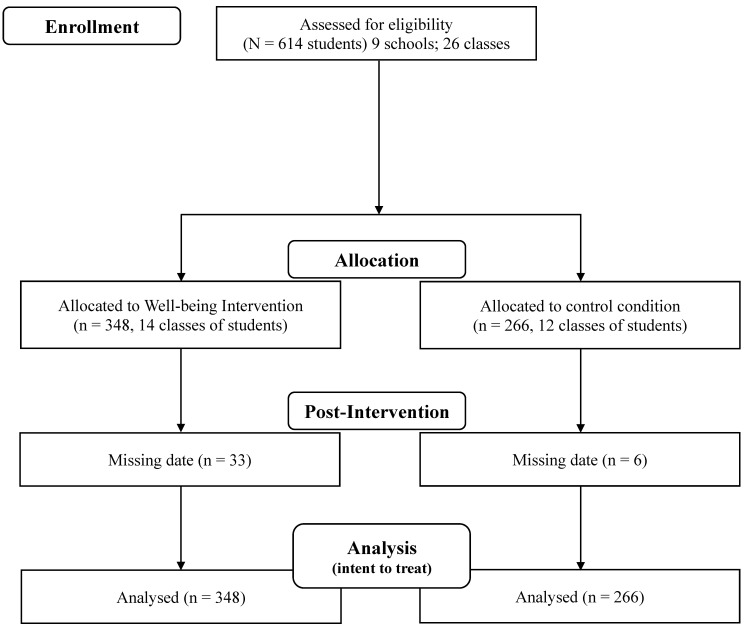
School intervention flowchart.

**Table 1 ijerph-20-06772-t001:** Sociodemographic and psychological characteristics of students.

	Pre-Intervention		Post-Intervention	
	Elementary Schoolchildren	Middle/High School Students	Elementary Schoolchildren	Middle/High School Students
	M (SD)	M (SD)	M (SD)	M (SD)
Gender (% female)	47.5%	53.3%	/	/
Age (range 8–17)	9.23 (0.50)	14.4 (0.67)	/	/
PWB Purpose in Life	15.21 (2.97)	14.58 (3.12)	15.17 (2.88)	14.45 (3.30)
SDQ overall stress	4.56 (4.74)	10.20 (6.53)		
SDQ emotional difficulties	1.26 (2.07)	2.77 (2.30)		
SDQ behavioral difficulties	0.87 (1.49)	1.75 (1.95)		
SDQ hyperactivity and attention difficulties	1.56 (1.99)	3.13 (2.91)		
SDQ difficulties getting alone with others	0.86 (1.15)	2.61 (1.86)		
SDQ kind and helpful behavior	8.13 (2.11)	5.97 (2.28)		
RCMAS Total Anxiety	11.11 (5.47)	9.93 (5.26)		
Depression (CTI; SQ-Dep)	18.01 (9.65)	5.23 (4.62)		
Somatization (CSI; SQ-Som)	19.93 (15.09)	5.67 (4.48)		

SDQ = Strengths and Difficulties Questionnaire; RCMAS = Revised Children’s Manifest Anxiety Scale; CTI = Cognitive Triad Inventory for Children; CSI = Children’s Somatization Inventory; SQ-Dep = Symptom Questionnaire-Depression Subscale; SQ-Som = Symptom Questionnaire-Somatization Subscale.

**Table 2 ijerph-20-06772-t002:** Regression models predicting purpose in life in the subsample of elementary schoolchildren (N = 158).

	Model 1		Model 2		Model 3		Model 4	
	β	*p*	β	*p*	β	*p*	β	*p*
Gender (1 = F; 2 = M)	0.065	0.407	−0.018	0.826	0.016	0.839	0.007	0.929
Age	0.025	0.754	−0.024	0.767	−0.018	0.817	−0.024	0.749
Wellbeing intervention (wb = 1; controls = 2)	−0.215	0.008	−0.206	**0.011**	−0.209	**0.008**	−0.174	**0.020**
SDQ overall stress			−1.981	0.401	−2.366	0.304	−1.738	0.421
SDQ emotional difficulties			0.969	0.347	1.149	0.253	0.810	0.391
SDQ behavioral difficulties			0.603	0.425	0.737	0.318	0.524	0.450
SDQ hyperactivity and attention difficulties			0.819	0.409	0.974	0.314	0.752	0.407
SDQ difficulties getting alone with others			0.292	0.605	0.436	0.430	0.331	0.522
SDQ kind and helpful behavior			0.192	0.064	0.164	0.105	0.121	0.204
RCMAS Total Anxiety					−0.237	**0.003**	0.064	0.518
Depression							−0.399	**<0.001**
Somatization							−0.113	0.181
R^2^	0.051		0.147		0.197		0.303	
R^2^ change	0.051		0.096		0.050		0.106	
F value	2.751	0.045	2.823	<0.004	3.595	<0.001	5.248	<0.001

SDQ = Strengths and Difficulties Questionnaire; RCMAS = Revised Children’s Manifest Anxiety Scale; wb = Wellbeing. Bold values are used to highlight significant predictors.

**Table 3 ijerph-20-06772-t003:** Regression models predicting purpose in life in the subsample of middle and high school students (N = 456).

	Model 1		Model 2		Model 3		Model 4	
	β	*p*	β	*p*	β	*p*	β	*p*
Age	−0.007	0.886	0.023	0.641	0.024	0.590	0.050	0.252
Gender (1 = F; 2 = M)	−0.020	0.673	**0.004**	0.928	−0.142	**0.003**	−0.140	**0.002**
Wellbeing intervention (wb = 1; controls = 2)	−0.067	0.161	−0.046	0.354	−0.082	0.067	−0.091	**0.037**
SDQ overall stress			0.206	0.636	−0.123	0.754	−0.057	0.884
SDQ emotional difficulties			−0.129	0.422	−0.022	0.878	−0.044	0.760
SDQ behavioral difficulties			0.248	0.129	0.277	0.061	0.227	0.117
SDQ hyperactivity and attention difficulties			−0.487	**0.019**	−0.187	0.323	−0.191	0.304
SDQ difficulties getting alone with others			−0.088	0.525	0.059	0.637	0.042	0.736
SDQ kind and helpful behavior			−0.095	0.100	−0.042	0.422	−0.029	0.562
RCMAS Total Anxiety					−0.448	**<0.001**	−0.245	**<0.001**
Depression							−0.317	**<0.001**
Somatization							0.039	0.442
R^2^	0.005		0.064		0.238		0.282	
R^2^ change	0.055		0.058		0.173		0.045	
F value	0.795	0.497	3.368	<0.001	13.783	<0.001	14.469	<0.001

SDQ = Strengths and Difficulties Questionnaire; RCMAS = Revised Children’s Manifest Anxiety Scale; wb = Wellbeing. Bold values are used to highlight significant predictors.

## Data Availability

Data are available from the authors following specific request to the corresponding author.

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
