# Peer review of "Improving Purpose in Life in School Settings"

_ijerph, 2023, doi:10.3390/ijerph20186772_

Round 1

Reviewer 1 Report

Thank you for the opportunity to review this manuscript, which describes a study examining purpose in life in school settings following a well-being intervention. My main feedback is around the methodology and statistical analysis. While, this is an interesting and worthwhile topic to investigate, there are some serious methodological issues that need to be addressed.

1. Given nested structure of data (students in classes in school), it is surprising a multilevel analysis approach was not employed (hierarchical linear modeling). Within and between class variance on outcome variable can then be partitioned and strengthen study results as variance in purpose in life lies between classes can be modeled as a function of class characteristics. Traditional regression approaches, like the authors used, assumes all observations are independent of one another. This assumption is violated with the nested study design, which is why HML should be used since it can analyze the variance of outcome at all levels.

2. Also, given clustered design, Intraclass correlation coefficient (ICC) should be computed and reported.

3. While recruitment and design procedures are cited as described in previous studies a bit more information in the current manuscript would be helpful. For instance, how many schools were recruited?, how many classes?, what was breakdown (n) of intervention and control?, where any schools/classes excluded and if so why?. A consort diagram would be beneficial as well. information about intervention, class topics/activities, and participant recruitment can be included as supplementary material.

4. Cronbach alpha of .139 for purpose in life scale is very low, however authors state psychometric properties are good. This needs to be changed and alpha value needs to be further looked into as it is highly unreliable.

5. Additionally, alpha was given for elementary and middle/high school sample for anxiety scale, but not for purpose in life or depression. Please be consistent. Also, SDQ does not have alpha reported, need for all subscales. Furthermore, please put how many questions are in each subscale (stated 24 items total in 5 subscales) and what point scale is used for the survey.

6. Was there any missing data? How was it handled if there was.

7. Why was baseline purpose in life not included in regression analysis? Scores for purpose in life dropped after the intervention in both elementary and middle/high school students and needs to explored and discussed further. What were pre and post purpose in life scores for control and intervention and was the trajectory the same for both conditions? It is very surprising that baseline mean scores for purpose in life were stated in results (lines 265-269) but not mentioned or explored further at all in the discussion. Additionally, the discussion is weak in general and very short (3 paragraphs minus limitations and conclusions). It needs to be expanded.

8. Table with F values, p values, effect size, mean, and standard deviation for pre and post intervention should be reported for variables: purpose in life, depression, anxiety, somatization, and SDQ subscales.

9. Was data collected on other dimensions of psychological well-being? It seems from prior articles cited by authors that it was. If so, why were the other dimensions not also included in the current analysis? Do variables (depressive symptoms, anxiety, SDQ) still predict post intervention purpose in life after controlling for pre intervention levels of the other dimensions of psychological well-being? Rationale for only using purpose in life is not clear with the introduction. Based on factor analysis all dimensions are highly correlated and connect therefore, weather just purpose in life is a predictor or other dimensions or to is not clear. Justification in the introduction is needed for exclusion of other dimensions from current study is needed if data is available and if data is not available, it needs to be included as a limitation that other dimensions could not be examined. 

Proof reading needs to be done carefully to catch typos and grammatical mistakes. Here are a few examples:

1. Line 50: replace Ruini et al. (Ruini et al., 2017) with Ruini and colleagues (2017)

2. Line 57: Delete names in parenthesis (Tavernier & Willoughby), should only have year since names are stated in sentence. 

3. Line 73: Delete name in parenthesis (Erickson), again only year should be there. 

4. Line 96: extra ')' needs to be deleted

5. Line 122: says 'purpose il life' - should be purpose in life

6. Line 124: two parenthesis ()() should be: (School WBT; Ruini et al., 2006,2009,2001; Tomba et al., 2010). 

7. Line 139: Write out single numbers. Not '4', but 'four'

8. Line 155: semicolon between years and mean

Author Response

Thank you for the opportunity to review this manuscript, which describes a study examining purpose in life in school settings following a well-being intervention. My main feedback is around the methodology and statistical analysis. While, this is an interesting and worthwhile topic to investigate, there are some serious methodological issues that need to be addressed.

  1. Given nested structure of data (students in classes in school), it is surprising a multilevel analysis approach was not employed (hierarchical linear modeling). Within and between class variance on outcome variable can then be partitioned and strengthen study results as variance in purpose in life lies between classes can be modeled as a function of class characteristics. Traditional regression approaches, like the authors used, assumes all observations are independent of one another. This assumption is violated with the nested study design, which is why HML should be used since it can analyze the variance of outcome at all levels.

Thank you for this comment. We have received the same criticisms when we have submitted the other research articles mentioned along the text (Ruini et al 2009; 2020; Tombe et al., 2010) . At that time we consulted our statistician in our Institution. After evaluating the characteristics of our sample, our statistician suggested that the HLM approach would not add relevant information, considering the sample involved in the present research and its internal homogeneity. Furthermore, the dependent variables of this research concerned subjective emotional states (i.e., purpose in life, depression, anxiety, somatization), which would not be necessarily influenced by school or class context. However, in the published articles we have already reported variance component analyses and intraclass correlation values (ICC) for RCMAS, CTI, PIL and CSI total scores. Additionally, in the present version of the article we have added data on baseline differences among the middle/high school classes (calculated using MANOVA), with classes as fixed factor and PIL , depression, anxiety and somatizations as dependent variables. These analyses revealed that there were no significant differences in the dependent variables among the various classes, with the exception of  anxiety levels , which differed in one middle class compared to one high school class. (see result section).

We have also underlined the homogeneity of the sample by indicating that the schools involved in the project shared similar sociodemographic and geographic characteristics (see methods section)

  1. Also, given clustered design, Intraclass correlation coefficient (ICC) should be computed and reported.

See previous comment. In previous published article we have reported the ICC correlation: values of the intraclass correlation ranged from 0.03 (PWB total score) to 0.05 (CTI and CSI) to 0.07 (RCMAS total score) and that the classes at baseline were not significantly different on all the dependent variables, but CTI, where 2 classes in the control condition presented significantly different total scores. These data are now reported in the method and result sections.

  1. While recruitment and design procedures are cited as described in previous studies a bit more information in the current manuscript would be helpful. For instance, how many schools were recruited?, how many classes?, what was breakdown (n) of intervention and control?, where any schools/classes excluded and if so why?. A consort diagram would be beneficial as well. information about intervention, class topics/activities, and participant recruitment can be included as supplementary material.

Thanks for this comment. Now we have added these details about recruitment and we have inserted a consort diagram (figure 1)

  1. Cronbach alpha of .139 for purpose in life scale is very low, however authors state psychometric properties are good. This needs to be changed and alpha value needs to be further looked into as it is highly unreliable.

The statement about good psychometric properties concerned the PWB questionnaire. Now we added some critical comments about the low value of the alpha value of PIL both in the assessment section and in the limitation section.

  1. Additionally, alpha was given for elementary and middle/high school sample for anxiety scale, but not for purpose in life or depression. Please be consistent. Also, SDQ does not have alpha reported, need for all subscales. Furthermore, please put how many questions are in each subscale (stated 24 items total in 5 subscales) and what point scale is used for the survey.

We have revised this part and provided all alpha values for the scales.

  1. Was there any missing data? How was it handled if there was.

Missing data are indicated in the consort diagram, They were handled with an intent-to-treat approach

  1. Why was baseline purpose in life not included in regression analysis? Scores for purpose in life dropped after the intervention in both elementary and middle/high school students and needs to explored and discussed further. What were pre and post purpose in life scores for control and intervention and was the trajectory the same for both conditions? It is very surprising that baseline mean scores for purpose in life were stated in results (lines 265-269) but not mentioned or explored further at all in the discussion. Additionally, the discussion is weak in general and very short (3 paragraphs minus limitations and conclusions). It needs to be expanded.

Pre-post differences in PIL were already calculated and reported in previously published papers (Ruini et la., 2009; 2020; Tomba et al 2010). In those investigations, a repeated measure design was used and it revealed significant changes from pre to post intervention in students assigned to the wellbeing intervention. In the present research we were interested to further explore the single dimension of PIL and its correlates. We added a paragraph at the end of the introduction section to better explain the main focus of the present investigation.

  1. Table with F values, p values, effect size, mean, and standard deviation for pre and post intervention should be reported for variables: purpose in life, depression, anxiety, somatization, and SDQ subscales.

We added a table with sociodemographic characteristics, pre-post intervention values for PIL and baseline mean score of the variables entered in the regression models (see table 1). As state in the previous comment, pre-post changes in well-being , depression,anxiety were already reported in previously published papers. 

  1. Was data collected on other dimensions of psychological well-being? It seems from prior articles cited by authors that it was. If so, why were the other dimensions not also included in the current analysis? Do variables (depressive symptoms, anxiety, SDQ) still predict post intervention purpose in life after controlling for pre intervention levels of the other dimensions of psychological well-being? Rationale for only using purpose in life is not clear with the introduction. Based on factor analysis all dimensions are highly correlated and connect therefore, weather just purpose in life is a predictor or other dimensions or to is not clear. Justification in the introduction is needed for exclusion of other dimensions from current study is needed if data is available and if data is not available, it needs to be included as a limitation that other dimensions could not be examined.

We added in the introduction and discussion section that the 6 dimensions of eudaimonic wellbeing are intercorrelated, and a large body of research already explored this issue. In the present research we focused on one specific dimension (the topic of this special issues) and we explored its correlates and predictors, including the beneficial effect of the school intervention

Reviewer 2 Report

The work proposed here contributes to further explore the topic related to Purpose in life from a developmental perspective, investigating the effectiveness of well-being promoting intervention in supporting this dimension. 

From the theoretical point of view, the citation apparatus is complete and exhaustive; however, in defining the construct related to eudemonia, it is suggested to cite additional works, which in the literature have helped to discriminate this construct from other related ones. In particular, it is appropriate to introduce reference to the works:

Huta, V., & Waterman, A. S. (2014). Eudaimonia and its distinction from hedonia: Developing a classification and terminology for understanding conceptual and operational definitions. Journal of happiness studies, 15, 1425-1456.

Deci, E. L., & Ryan, R. M. (2008). Hedonia, eudaimonia, and well-being: An introduction. Journal of happiness studies, 9, 1-11.

It is also suggested that the citation apparatus be further diversified by containing the number of self-citational references.

From the methodological point of view, in defining the sample, it is necessary to define more precisely what is meant by cases of "severe disability," including using examples. Indeed, it is not entirely clear which types of disabilities were excluded from the sampling; moreover, it is necessary to justify their exclusion from a scientific point of view.

In addition, it is necessary to supplement Ryff's Psychological Well-being presentation with data on the reliability.

Author Response

the work proposed here contributes to further explore the topic related to Purpose in life from a developmental perspective, investigating the effectiveness of well-being promoting intervention in supporting this dimension. 

From the theoretical point of view, the citation apparatus is complete and exhaustive; however, in defining the construct related to eudemonia, it is suggested to cite additional works, which in the literature have helped to discriminate this construct from other related ones. In particular, it is appropriate to introduce reference to the works:

Huta, V., & Waterman, A. S. (2014). Eudaimonia and its distinction from hedonia: Developing a classification and terminology for understanding conceptual and operational definitions. Journal of happiness studies, 15, 1425-1456.

Deci, E. L., & Ryan, R. M. (2008). Hedonia, eudaimonia, and well-being: An introduction. Journal of happiness studies, 9, 1-11.

We have now added these articles in the introduction and discussion section.

It is also suggested that the citation apparatus be further diversified by containing the number of self-citational references.

We have delated some citations, however, this work is strongly related to other previously published articles and for this reason it was not possible to remove them form the reference list.

From the methodological point of view, in defining the sample, it is necessary to define more precisely what is meant by cases of "severe disability," including using examples. Indeed, it is not entirely clear which types of disabilities were excluded from the sampling; moreover, it is necessary to justify their exclusion from a scientific point of view.

We have added this sentence in the methods section: Severe disability (such as neurodevelopmental disorder, autism spectrum disorder, visual impairments) or impossibility to speak and understand the Italian language were considered exclusion criteria since children could not be properly engaged in the activities and could not provide their individual contribution.

In addition, it is necessary to supplement Ryff's Psychological Well-being presentation with data on the reliability.

We have expanded the part on the psychometric characteristics of the PWB and added comments in the assessment and limitation section.

Reviewer 3 Report

The current study examines whether a short 4-session well-being intervention predicts purpose of life, controlling for demographic indicators, anxiety, depression, somatization, and teacher-rated strengths and difficulties among middle and high school students as well as elementary school students. The following are offered as points of clarity to further strengthen the submission:

1.       Consider adding random assignment of schools to the Methods section of the abstract;

2.       Consider adding an additional table (a new Table 1) in section “2.1 Sample” or “2.3 Data Analysis” of descriptive statistics for the full sample and the high/middle school and elementary school samples separately so the reader understands the sample(s) better/at a glance.  

3.       The alpha for the purpose in life subscale is reported as .139 on page 4. This is extremely low (obviously) and deserves discussion/justification of the repercussions of this, as purpose in life is the dependent measure.

4.       There is mention throughout the manuscript of changes in purpose of life based on intervention (ex. “The purpose of this investigation was to explore...and to verify it was amenable of change following an eudaimonic well-being intervention”, p. 7). However, the current study does not appear to have controlled for initial/baseline purpose in life or assessed within individual change. Clarification would help, or adjust the “change” language to indicate the study merely predicted purpose in life, but cannot really assert purpose in life had anything to do with the intervention without any indication that groups didn’t differ initially (the descriptives table recommended above would help, if it also had comparisons of means testing).

5.       Section 3 Results should include in the text the direction of the effect as opposed to mentioning which measures were predictive. For instance, higher depression predicted lower purpose in life.

Author Response

The current study examines whether a short 4-session well-being intervention predicts purpose of life, controlling for demographic indicators, anxiety, depression, somatization, and teacher-rated strengths and difficulties among middle and high school students as well as elementary school students. The following are offered as points of clarity to further strengthen the submission:

  1. Consider adding random assignment of schools to the Methods section of the abstract; Thank you for this comment. We added the randomization in the abstract.
  2. Consider adding an additional table (a new Table 1) in section “2.1 Sample” or “2.3 Data Analysis” of descriptive statistics for the full sample and the high/middle school and elementary school samples separately so the reader understands the sample(s) better/at a glance.  

We added a table with sociodemographic characteristics, pre-post intervention values for PIL and baseline mean score of the variables entered in the regression models (see table 1) Pre-post changes in well-being , depression,anxiety were already reported in previously published papers. 

  1. The alpha for the purpose in life subscale is reported as .139 on page 4. This is extremely low (obviously) and deserves discussion/justification of the repercussions of this, as purpose in life is the dependent measure. Now we added some critical comments about the low value of the alpha value of PIL both in the assessment section and in the limitation section.
  2. There is mention throughout the manuscript of changes in purpose of life based on intervention (ex. “The purpose of this investigation was to explore...and to verify it was amenable of change following an eudaimonic well-being intervention”, p. 7). However, the current study does not appear to have controlled for initial/baseline purpose in life or assessed within individual change. Clarification would help, or adjust the “change” language to indicate the study merely predicted purpose in life, but cannot really assert purpose in life had anything to do with the intervention without any indication that groups didn’t differ initially (the descriptives table recommended above would help, if it also had comparisons of means testing). Pre-post differences in PIL were already calculated and reported in previously published papers (Ruini et la., 2009; 2020; Tomba et al 2010). In those investigations, a repeated measure design was used and it revealed significant changes from pre to post intervention in students assigned to the wellbeing intervention. In the present research we were interested to further explore the single dimension of PIL and its correlates. We added a paragraph at the end of the introduction section to better explain the main focus of the present investigation
  3. Section 3 Results should include in the text the direction of the effect as opposed to mentioning which measures were predictive. For instance, higher depression predicted lower purpose in life. We have now clarified that lower levels of depression and anxiety were correlated with higher purpose in life.

Round 2

Reviewer 1 Report

The authors edits are sufficient and I have no other issues. 

Author Response

Thank you for your positive answer

Reviewer 3 Report

This reviewer appreciates the detailed responses incorporated into the manuscript to address the initial concerns. The revised manuscript has been strengthened and stands independently from other prior published studies referenced within.  

no comments/no significant issues.

Author Response

Thank you for your positive answer